# *Colletotrichum galinsogae* sp. nov. Anthracnose Pathogen of *Galinsoga parviflora*

Igor Kazartsev * , Maria Gomzhina , Elena Gasich and Philipp Gannibal

Laboratory of Mycology and Phytopathology, All-Russian Institute of Plant Protection (FSBSI VIZR), Pushkin, 196608 St. Petersburg, Russia; egasich@vizr.spb.ru (E.G.); fgannibal@vizr.spb.ru (P.G.)

* Correspondence: ikazartsev@vizr.spb.ru

**Abstract:** *Galinsoga parviflora* is an herbaceous dicotyledonous plant in the *Asteraceae* that is common in disturbed habitats and agricultural areas across various temperate and subtropical regions of the world. In this study, several pathogenic strains were isolated from this host, and further morphological and phylogenetic analysis of DNA sequences of nuclear rDNA ITS1-5.8S-ITS2 (ITS barcode) and five other gene regions (*act*, *chs-1*, *gapdh*, *his3*, and *tub2*) revealed a new species, described here as *Colletotrichum galinsogae* sp. nov. The pathogenicity of *C. galinsogae* sp. nov. was also tested and confirmed on leaf segments and seedlings of *G. parviflora*.

**Keywords:** *Colletotrichum*; *C. destructivum* species complex; *Galinsoga parviflora*; multilocus analysis; species nova



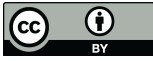

## 1. Introduction

*Galinsoga parviflora* Cav. (common names include gallant soldier, quickweed, and potato weed) is an herbaceous dicotyledonous plant from the family *Asteraceae*. It is common in disturbed habitats and agricultural areas across various temperate and subtropical regions of the world [1]. The native range of *G. parviflora* is considered to be the mountainous area of Central America [2]. It was introduced to Europe at the end of the eighteenth century and eventually spread to many European countries. Nowadays, *G. parviflora* is a cosmopolitan annual weed that is particularly common in Eastern Asia, Africa, Australia, and the Pacific Islands. This weed is also recorded in the territories of the former USSR [3]. Despite being classified as a weed, *G. parviflora* is non-toxic and can be used as food and fodder [1]. Also, extracts and compounds isolated from this plant possess a number of promising biological activities [4]. A significant number of fungal and fungal-like pathogens have been isolated from *Galinsoga* spp., including reports of fungi from the genus *Colletotrichum* Corda [5,6].

*Colletotrichum* (*Sordariomycetes*, *Glomerellales*, and *Glomerellaceae*) is one of the most common and important genera of plant-pathogenic fungi that can cause reduced crop yields and quality and significant agricultural losses. In this regard, the range of the studied *Colletotrichum* host plants is skewed towards agricultural crops, as wild plants are less often the subject of phytopathological investigations despite representing a rich source of pathogen biodiversity. Some members of *Colletotrichum* and their phytotoxins are considered important biological control agents for weeds; some *Colletotrichum* strains are registered as mycoherbicides [7].

*Colletotrichum* includes cryptic species, and the delimitation of species based solely on morphological features and host plants is an extremely challenging task. Recently, the taxonomy of this genus has undergone significant changes, including the description of a new species and the removal of some old ones. This became possible mainly through the clarification of their phylogenetic status by molecular genetic methods.

The aim of this study was to identify *Colletotrichum* sp. isolated from *G. parviflora* with anthracnose symptoms collected in Eastern Europe by phylogenetic, micromorphological, and cultural features. An additional goal was to validate the pathogenicity of the identified strains.

## 2. Materials and Methods

### 2.1. Sample Collection and Fungal Isolation

Three specimens of *G. parviflora* with leaves and stems exhibiting anthracnose symptoms were collected in different locations of Suma Oblast (Ukraine) in 1995 and 2013, and in Pskov Oblast (Russia) in 2013 (Table A1). For fungal isolation, infected tissues were surface sterilized with silver nitrate (1%) and then rinsed three times with sterile water and incubated on potato sucrose agar (PSA) [8]. Three strains, one from each sample, were previously identified as *Colletotrichum* sp. and stored in microtubes on PSA at 4 °C in the mycological collection of pure cultures in the Laboratory of Mycology and Phytopathology (MF; All-Russian Institute of Plant Protection (VIZR), Saint Petersburg, Russia).

### 2.2. DNA Extraction, PCR and Sequencing

For DNA isolation, the test strains were grown on PSA at 24 °C for 2 weeks. Then, fungal mycelia were collected from the surface of the medium and placed in a 2 mL microcentrifuge tube to be processed using the CTAB DNA extraction protocol [9]. The internal transcribed spacer (ITS) regions, including 5.8S rDNA, were amplified with primers ITS1/ITS4 [10]. Partial gene sequences of the actin (*act*), chitin synthase 1 (*chs-1*), intron of the glyceraldehyde-3-phosphate dehydrogenase (*gapdh*), histone H3 (*his3*), and beta-tubulin (*tub2*) genes were amplified with primers Act512F/Act783R [11], CHS-354R/CHS79F [11], GDF1/GDR1 [12], CYLH3F/CYLH3R [13], and btub2Fw/btub4Rd [14], respectively. The amplification reactions had a total reaction volume of 25 μL, including dNTPs (200 μM), forward and reverse primers (0.5 μM each), Taq DNA polymerase (1 U; Qiagen, Hilden, Germany), 10× PCR buffer with $Mg^{2+}$ and $NH^{4+}$ ions, and 1–10 ng of total genomic DNA. The parameters for DNA amplification were set according to the specifications of each primer pair.

After electrophoresis on a 1% agarose gel stained with ethidium bromide, the bands chosen for sequencing were excised and purified with silica particles [15]. Then, the amplicons were sequenced by Sanger's method [16] on an ABI Prism 3500 analyzer (Applied Biosystems, Thermo Fisher Scientific, Waltham, MA, USA) with a Bigdye Terminator v3.1 Cycle Sequencing Kit (Applied Biosystems) according to the manufacturer's instructions.

### 2.3. Phylogenetic Studies

The nucleotide sequences were checked and manually edited using Vector NTI Advance 11.5.1 software (Life Technologies, Thermo Fisher Scientific). Final consensus sequences of target strains were subjected to a BLASTn search in GenBank (https://blast.ncbi.nlm.nih.gov/Blast.cgi (accessed on 30 January 2023)) to identify the closest species complex [17]. The new sequences and reference sequences retrieved from GenBank (Table S1) were aligned using the MUSCLE algorithm [18] performed in MEGA X [19]. SequenceMatrix 1.7.3 [20] was used to concatenate loci and obtain a dataset for further multilocus analysis with Bayesian inference (BI) based on the Markov Monte Carlo (MCMC) chain algorithm and maximum likelihood (ML) analysis. The appropriate nucleotide (nt) substitution models for each locus were determined using jmodeltest 2.1.10 [21] based on the Bayesian information criterion. BI phylogenetic analysis for the concatenated dataset was performed in MrBayes v. 3.2.6 [22] by running $10^7$ generations with two independent runs and sampling every 500 generations. The first 25% of the generations of MCMC trees were discarded as burnin and posterior probabilities were calculated. The 50% majority rule consensus tree was visualized in FigTree 1.4.4. ML phylogenetic reconstruction for the concatenated dataset was conducted in IQ-tree 1.6.12 with 10,000 ultrafast bootstrap approximations [23–25].

Retrieved phylogenetic data were visualized in MEGA X. The new sequences obtained were deposited in GenBank (https://www.ncbi.nlm.nih.gov (accessed on 12 April 2023)).

### 2.4. Morphology

Pure cultures of the studied strains were incubated on PSA, oatmeal agar (OA) [26], and synthetic nutrient-poor agar medium (SNA) [27]. To enhance sporulation, autoclaved filter paper and double-autoclaved stems of *Anthriscus sylvestris* (L.) Hoffm. were placed onto the surface of SNA in Petri dishes. These Petri dishes were kept for 7 d in darkness and then for 7 d under a 12:12 h L:D photoperiod with near-ultraviolet light to stimulate sporulation. Colony diameter and morphology were examined after 7 and 14 d, respectively. Spore-bearing structures were estimated after 14 d. Appressoria were obtained and observed in germinated spores in a drop of sterile water after 4 d. For each strain, 50 conidia were observed and measured with an Olympus SZX16 stereomicroscope (Olympus Corp., Tokyo, Japan) and an Olympus BX53 microscope. Images were captured with a Prokyon camera (Jenoptik AG, Jena, Germany) with Nomarski differential interference contrast.

### 2.5. Pathogenicity Test

To confirm the pathogenicity of the strains, leaf segments of *G. parviflora* were inoculated with mycelial suspension. For inoculation, strains were grown on liquid soybean nutrient medium with the following composition: $KH_2PO_4$ (0.2%), $(NH_4)_2SO_4$ (0.1%), $MgSO_4$ (0.1%), glucose (2%), and soy flour (1%). Fifty milliliter of medium in 250 mL flasks were inoculated with three 5 mm mycelial discs, cut from 2-week-old colonies grown on PSA, and kept on an orbital shaker (200 rpm) for 4 d at 24 °C. The mycelium was separated from the liquid culture by filtering, then dried and ground. This resultant powder was used to prepare a mycelial suspension of 100 mg/mL using sterile water. Leaf segments (1.5 × 1.5 cm) were placed in Petri dishes premoistened with sterile water filter paper to create a moisture chamber effect. A drop of mycelial suspension (10 μL) was applied to the center of the previously injured or intact leaf segments on the abaxial or adaxial surface. In the control, the wounded or non-wounded leaf segments were treated with 10 μL of sterile distilled water. After inoculation, closed Petri dishes with leaf segments were kept at room temperature and evaluated for the presence of symptoms at 4, 7, and 14 d post-treatment (dpt).

Further pathogenicity tests were conducted with the inoculation of *G. parviflora* seedlings at the second true leaf stage. Mycelial suspension was prepared as described above. Plants were sprayed with mycelial suspension at 50 mg/mL and placed in wet chambers for 24 or 48 h (dew period). The plants were transferred to a greenhouse and grown under ambient conditions. At 2 and 7 dpt, the leaves of each plant treated with suspension were counted and individually rated for disease symptoms using a 6-point scale (0, no disease; 1, 0–5%; 2, 6–25%; 3, 26–75%; 4, 76–95%; 5, >95% of leaf surface with necrosis; and 6, leaf dead completely). The proportion of necrotic leaf area was calculated using the formula $(2.5 \times n_1 + 15 \times n_2 + 50 \times n_3 + 85 \times n_4 + 97.5 \times n_5 + 100 \times n_6)/N$, where $n_x$ is the number of leaves with rating x and N is the total number of leaves treated [28]. Experiments with leaf segments and seedlings were conducted with five replicates of each fungal strain. Subsequent reisolation of pathogens from the inoculated leaf segments and seedlings and identification were made to fulfill Koch's postulates.

## 3. Results

### 3.1. Phylogenetic Analyses

Searching for the closest match with the BLASTn algorithm showed that all sequences of the test strains could be preliminarily considered to be in the *Colletotrichum destructivum* O'Gara species complex. For the BI and ML multilocus phylogenetic analyses of the *C. destructivum* species complex, sequences of 26 *Colletotrichum* ex-type and reference strains were downloaded from GenBank. *Colletotrichum gloeosporioides* (Penz.) Penz. & Sacc. IMI356878 was used as an outgroup (Table S1). A multilocus concatenated dataset

comprised 2318 characters, including alignment gaps and missing data. Locus boundaries were: *act*, 1–248 nt; *chs-1*, 249–511 nt; *his3*, 512–906 nt; ITS, 907–1467 nt; *tub2*, 1468–1960 nt; and *gapdh*, 1961–2318 nt. The best substitutional models for BI and ML were K80 + G for *act*, *chs-1*, and *tub2*, HKY + G for *his3*, K80 + G + I for ITS, and HKY + I for *gapdh*. The consensus tree obtained from ML analysis of the multilocus alignment grouped strains MF-13.2, MF-13.25, and MF-13.27 in a distinct monophyletic clade with a bootstrap value of 100 (Figure 1) that did not include any ex-type or representative *Colletotrichum* strains and is considered to represent a new *Colletotrichum* species. The topologies of the BI trees calculated from multilocus datasets were consistent with the results from the ML analysis.

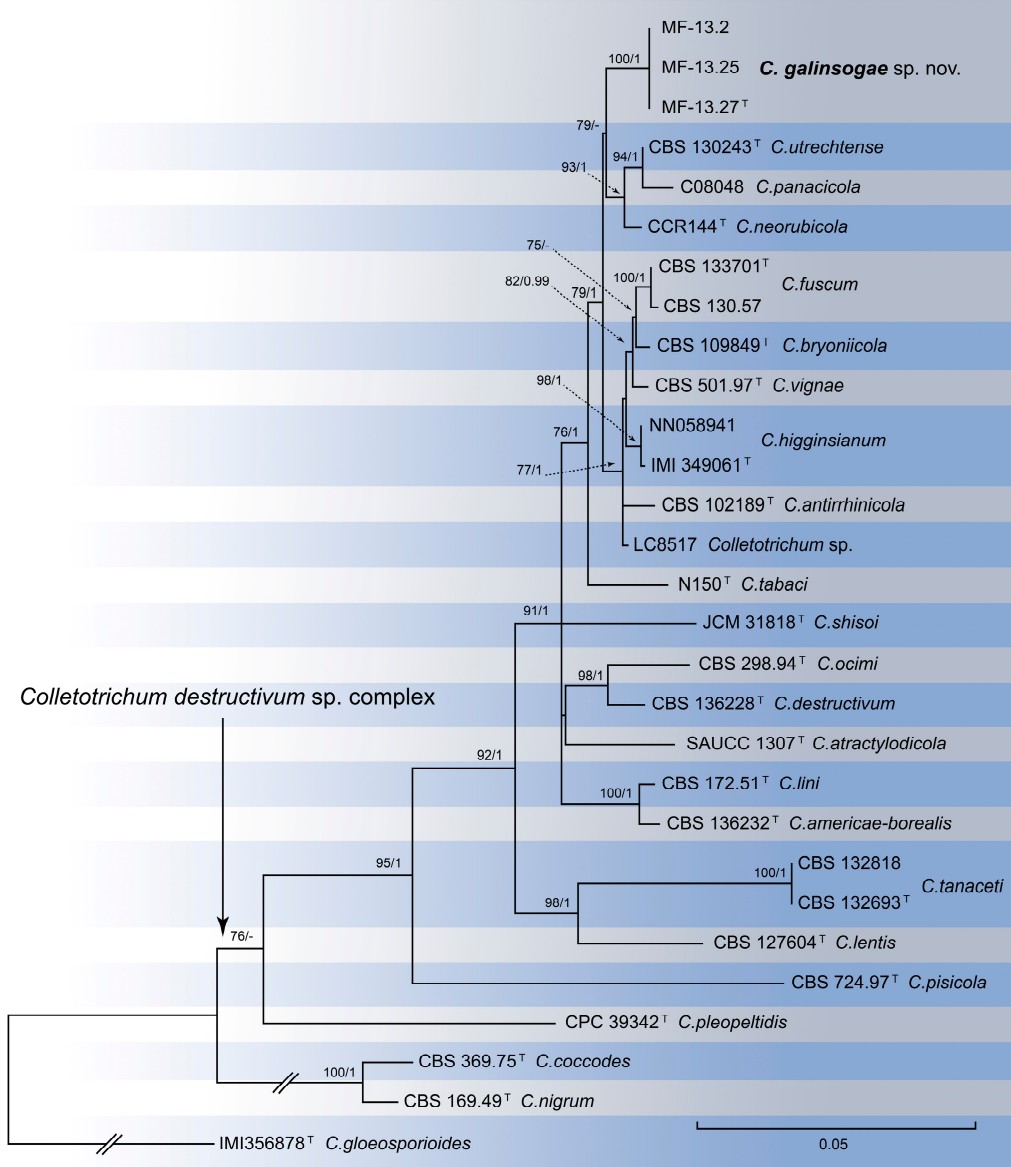

**Figure 1.** Phylogenetic tree derived from maximum likelihood (ML) analysis of the combined *act*, *chs-1*, *gapdh*, *his3*, ITS, *tub2* sequences of *C. destructivum* species complex, *C. coccodes*, and *C. nigrum*, with *C. gloeosporioides* IMI356878 used as outgroup. Numbers above the branches are respective ML bootstraps (**left**, MLBS ≥ 75%) and Bayesian posterior probabilities (**right**, BPP ≥ 0.95). *Colletotrichum galinsogae* sp. nov. strains (MF 13.2, MF 13.25, MF 13.27) are highlighted in bold. The scale bar indicates the number of expected changes in the nucleotide sequence of each 100 bp. Branches with diagonal double bars are truncated fivefold. Ex-type strains are indicated with superscript letter "T".

### 3.2. Taxonomy

Based on the multilocus analysis, the three *Colletotrichum* strains MF-13.2, MF-13.25, and MF-13.27 from *G. parviflora* were found to be distinct from any described species in the *C. destructivum* species complex and are, therefore, described here as a new species below.

*Colletotrichum galinsogae* Gasich, Gomzhina & Kazartsev, sp. nov. MycoBank MB#849474. Figures 2 and 3.

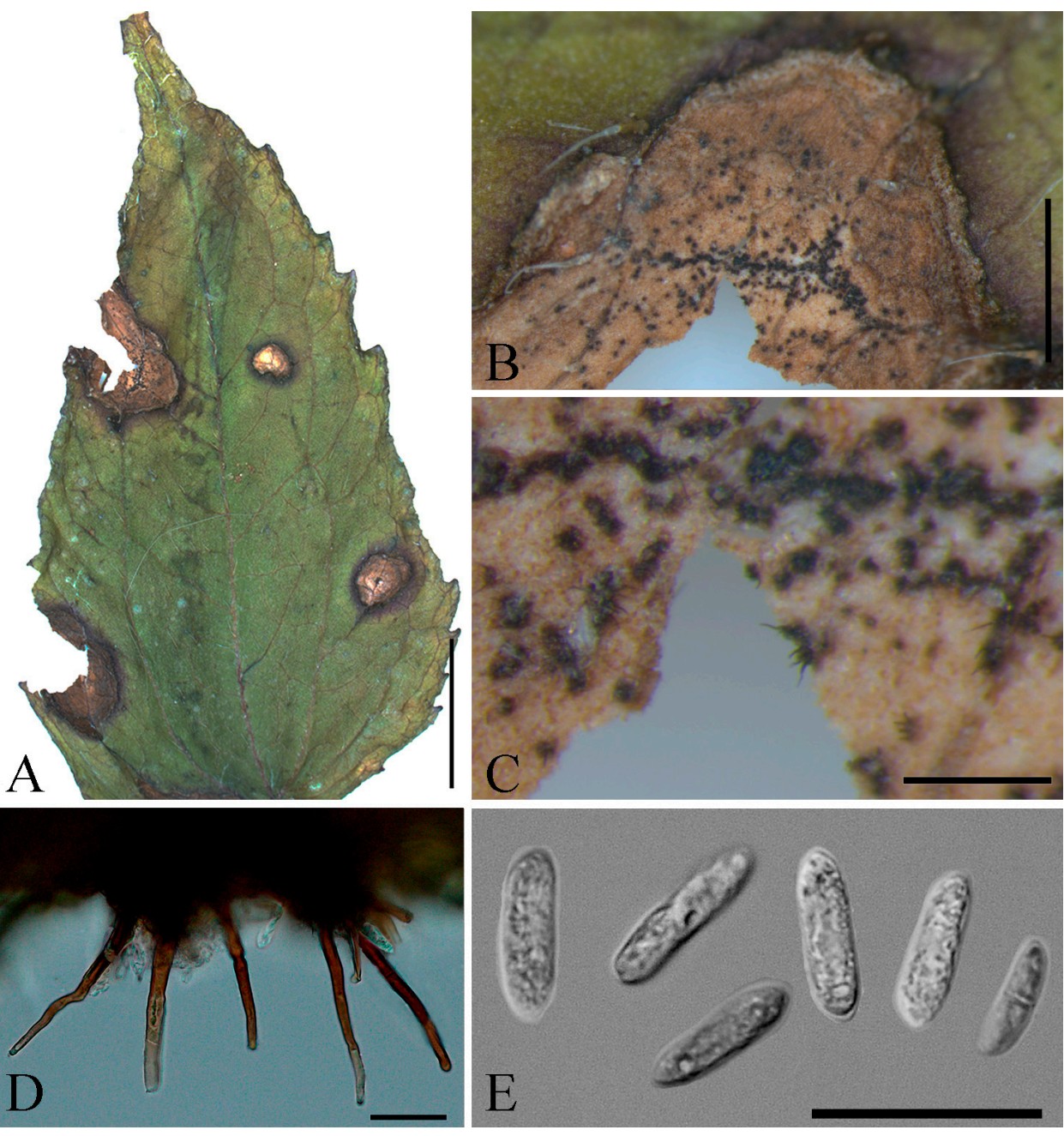

**Figure 2.** Holotype *Colletotrichum galinsogae* sp. nov., herbarium specimen LEP 132919. (**A**). Leaf of *Galinsoga parviflora* with leaf spots. (**B,C**). Acervuli in leaf of *Galinsoga parviflora*. (**D**). Setae. (**E**). Conidia. Scale bars: (**A**), 5 mm; (**B**), 1 mm; (**C**), 200 μm; and (**D,E**), 20 μm.

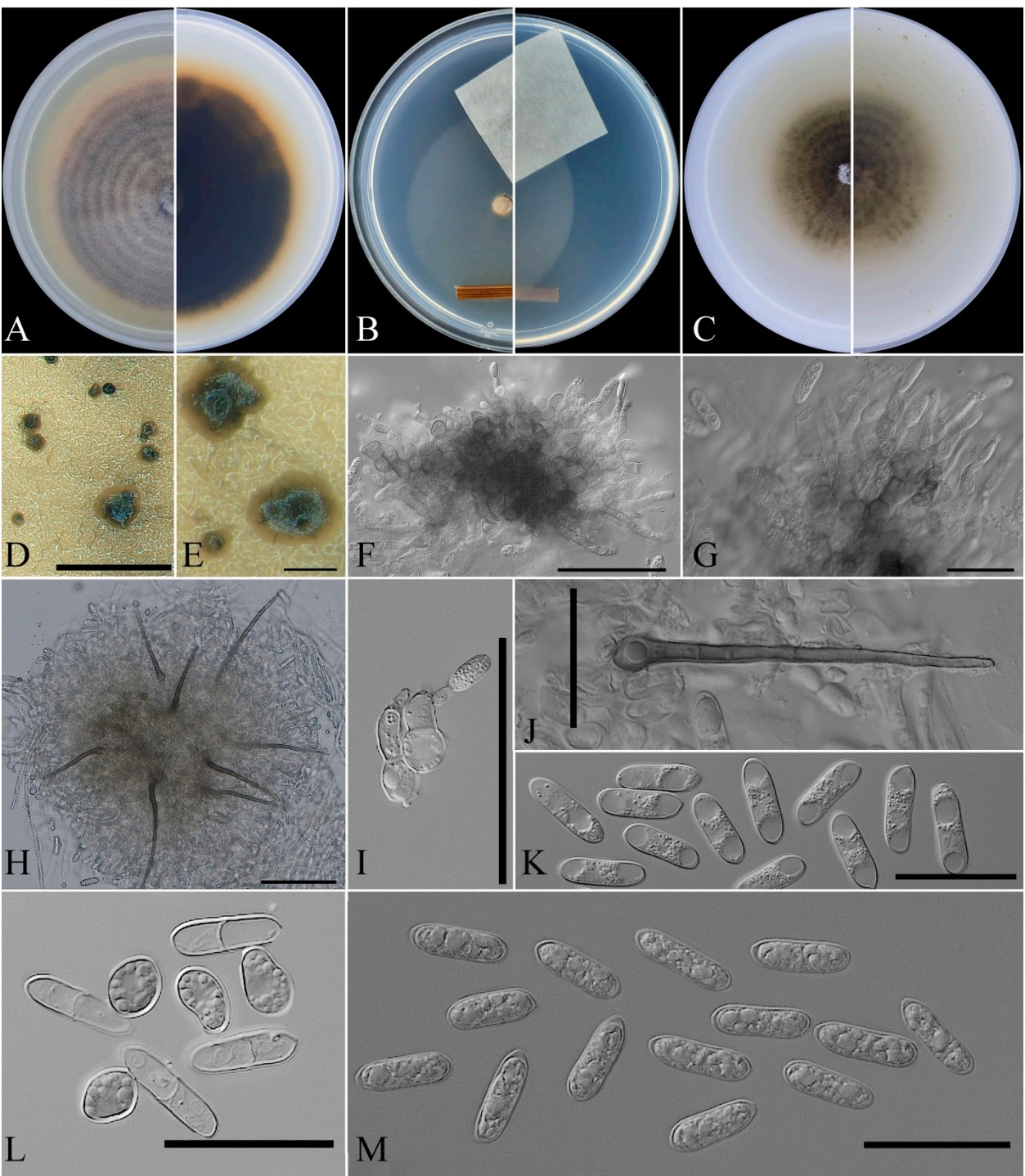

**Figure 3.** *Colletotrichum galinsogae* sp. nov. (ex-holotype culture MF-13.27) 14 d of growth. (**A**). Cultures on PSA, front (**left**) and reverse (**right**). (**B**). Cultures on SNA, front (**left**) and reverse (**right**). (**C**). Cultures on OA, front (**left**) and reverse (**right**). (**D–F,H**). Acervuli from OA. (**G,I**). Conidiophores from OA. (**J**). Seta from OA. (**K**). Conidia from *Anthriscus* stem. (**L**). Appressoria from germinated spores in a drop of sterile water after 4 d. (**M**). Conidia from OA. Scale bars: (**D**), 1 mm; (**E**), 200 μm; (**F,H,I**), 50 μm; and (**G,J–M**), 20 μm.

*Typification*: RUSSIA, PSKOV OBLAST: Velikie Luki district, Maykino, in leaves of *Galinsoga parviflora*, 09 August 2013, Ph. B. Gannibal (holotype LEP 132919 and ex-holotype living culture MF-13.27). GenBank (DNA sequences derived from the type: ITS OL647887, *act* OL676714, *chs-1* OQ815879, *gapdh* OQ815882, *his3* OQ815885, and *tub2* OL676749).

*Additional material examined*: UKRAINE, SUMY OBLAST: Seredina-Buda district, Golubovka, in stems of *G. parviflora*, 17 Aug 1995, E.L. Gasich (paratype LEP 135112 and ex-paratype living culture MF-13.2). GenBank (DNA sequences derived from the type: ITS OL647879, *act* OL676706, *chs-1* OQ815877, *gapdh* OQ815880, *his3* OQ815883, and *tub2* OL676741). UKRAINE, SUMY OBLAST: Seredina-Buda district, Seredina-Buda, in stems of *G. parviflora*, 24 July 2013, E.L. Gasich (paratype LEP 132984 and ex-paratype living culture MF-13.25). GenBank (DNA sequences derived from the type: ITS OL647885, *act* OL676712, *chs-1* OQ815878, *gapdh* OQ815881, *his3* OQ815884, and *tub2* OL676747).

Etymology: Named after the host plant genus *Galinsoga*, from which strains of this species are known.

Sexual morph not observed.

Asexual morph on OA. *Vegetative hyphae*, 2.1–5.4 μm diam, hyaline to pale brown, smooth-walled, septate, branched. *Chlamydospores* not observed. *Sclerotia* form in old culture, irregular form, immersed, black, numerous. *Conidiomata* acervular, conidiophores and setae formed on pale brown basal cells 3.5–6.9 μm diam. *Setae* not abundant, 0–8 per acervulus, dark brown, basal cell often paler, smooth-walled, 0–3 septate, 42–80 μm long, base conical, inflated, 5.2–6.6 μm wide, tip somewhat acute. Production of setae ceases during transfers and long-term storage, resumes in strains that were reisolated after artificial infection of the *G. parviflora*. *Conidiophores* hyaline, smooth-walled, to 32 μm long. *Conidiogenous cells* hyaline, cylindrical, smooth-walled, 3.6–5.3 × 2.3–2.7 μm. *Conidia* hyaline, smooth-walled, aseptate, straight, cylindrical with one end round and one end somewhat acute, 12.1–16.1 (13.9 ± 0.11) × 3.7–5.5 (4.4 ± 0.06) μm.

Asexual morph on SNA. *Vegetative hyphae* 3.1–5.8 μm diam, hyaline to pale brown, smooth-walled, septate, branched. *Chlamydospores* not observed. *Conidiomata* not developed, conidiophores and setae formed directly on hyphae or on plexus of hyphae. *Setae* not developed. *Appressoria* not observed. *Conidiophores* hyaline, smooth-walled, to 35 μm long. *Conidiogenous cells* hyaline, cylindrical, smooth-walled, 3.6–4.4 × 2.3–2.6 μm. *Conidia* hyaline, smooth-walled, aseptate, straight, cylindrical with one end round and one end somewhat acute, 13.8–18.3 (16.1 ± 0.12) × 3.4–5.3 (4.5 ± 0.05) μm.

Asexual morph on Anthriscus stem. *Vegetative hyphae* 2.3–3.8 μm diam, hyaline to pale brown, smooth-walled, septate, branched. *Chlamydospores* not observed. *Conidiomata* not developed, conidiophores and setae formed directly on hyphae. *Setae* very rare, dark brown, basal cell smooth-walled, 1–3 septate, 40–60 μm long, base cylindrical to conical, sometimes inflated, 4.9–5.3 μm wide, tip somewhat acute. *Conidiophores* hyaline, septate, branched, to 20 μm long. *Conidiogenous cells* hyaline, smooth-walled, cylindrical, 3.9–6.7 × 2–3.9 μm. *Conidia* hyaline, smooth-walled, aseptate, straight, cylindrical with one end round and one end somewhat acute, 11.0–16.6 (14.4 ± 0.16) × 4.0–5.2 (4.6 ± 0.04) μm.

Appressoria do not form in pure cultures on artificial media. In drop of sterile water after 4 d abundant, single, pale brown, smooth-walled, subglobose to elliptical, the edge entire to undulate, 5.6–9.2 (7.5 ± 0.12) × 3.6–6.8 (5.8 ± 0.08) μm.

Culture characteristics: Colonies on SNA flat with entire margin, very light gray to hyaline, aerial mycelium scarce, filter paper, and on *Anthriscus* stem partly covered with acervuli appearing as tiny gray spots, growth rate 17.5–22.5 mm in 7 d (32.5–36 mm in 10 d).

Colonies on OA flat with entire margin, covered with scattered acervuli, in old culture numerous immersed sclerotia, aerial mycelium scarce, light gray, surface from brown green in the center to beige gray near the margin, reverse from gray umber in the center to beige gray near the margin, growth rate 18–26 mm in 7 d (30.0–37.5 mm in 10 d). Conidia in mass light brown to hyaline.

Colonies on PSA felty-velvet, margin regular, covered with abundant aerial mycelium contains scattered black acervuli, seems striped due to concentric zones of lighter (dusty gray) and darker (bluish gray) color, entire colony has light purple tint, reverse night blue, light brown near the margin, growth rate 18–26 mm in 7 d (30.0–37.5 mm in 10 d).

Notes: during preservation in collection and numerous passages, strains lose their ability to produce setae in acervuli. This ability could be revived in strains after inoculation of host plants and subsequent reisolation.

### 3.3. Pathogenicity

The pathogenicity of *C. galinsogae* sp. nov. was tested using *G. parviflora* leaf segments and seedlings (Tables 1 and 2). The initial significant symptoms were recorded at 7 dpt on the adaxial sides of the leaf segments. The abaxial sides were less susceptible to infection, and symptoms were recorded only at 14 dpt. Overall, the infection of both wounded and intact leaf segments yielded similar results.

**Table 1.** Pathogenicity of *Colletotrichum galinsogae* sp. nov. strains on leaf segments of *Galinsoga parviflora*.

| Strain | Diameter of Necrosis, mm (4, 7 and 14 dpt *) | | | |
| --- | --- | --- | --- | --- |
| | Adaxial Side | | Abaxial Side | |
| | Not Wounded | Wounded | Not Wounded | Wounded |
| MF-13.2 | 0; 4.6 ± 0.7 **; 7.2 ± 1.1 | 0; 4.0 ± 0.6; 7.1 ± 0.4 | 0;0; 1.8 ± 0.2 | 0; 0; 3.2 ± 0.8 |
| MF-13.25 | 0; 2.6 ± 1,2; 9.8 ± 0.5 | 0; 5.1 ± 0.4; 10.1 ± 1.1 | 0; 0; 2.2 ± 0.6 | 0; 0; 2.8 ± 0.2 |
| MF-13.27 | 0; 5.3 ± 0.4; 11.2 ± 0.8 | 0; 3.5 ± 0.4; 7.9 ± 1.6 | 0; 0; 3.5 ± 1.4 | 0; <1; 4.1 ± 1.7 |

* dpt, days post treatment; ** values are means ± SEM.

**Table 2.** Pathogenicity test of *Colletotrichum galinsogae* sp. nov. strains on seedlings of *Galinsoga parviflora*.

| Strain | Necrotic Leaf Area, % | | | |
| --- | --- | --- | --- | --- |
| | 2 dpt * | | 7 dpt | |
| | dp ** 24 h | dp 48 h | dp 24 h | dp 48 h |
| MF-13.2 | 78.2 ± 7.0 *** | 72.9 ± 7.0 | 100 | 100a |
| MF-13.25 | 82.3 ± 11.8 | 93.5 ± 4.7 | 100 | 100 |
| MF-13.27 | 62.7 ± 11.5 | 95.3 ± 4.6 | 100 | 100 |

* dpt, days post-treatment; ** dp, dew period; *** values are means ± SEM.

The inoculated *G. parviflora* seedlings proved susceptible to *C. galinsogae* sp. nov. in both tested variants, with dew periods lasting for 24 and 48 h. In most cases, at 2 dpt, necrotic leaf area exceeded 60%, and after 7 dpt, it reached 100%.

### 4. Discussion

The newly discovered species, *C. galinsogae* sp. nov., was isolated from the leaves and stems of *G. parviflora*, which originated in the Northern regions of Ukraine and Northern European Russia. The new species was described based on macro- and micro-morphological characters as well as multilocus phylogenetic analysis of *act*, *chs-1*, *gapdh*, *his3*, ITS, and *tub2* sequences. *Colletotrichum galinsogae* sp. nov. forms a highly supported monophyletic group within the *C. destructivum* species complex and is phylogenetically close to the clade containing three species: *Colletotrichum neorubicola* Yu Li, J. Gao & L.P. Liu; *Colletotrichum utrechtense* Damm; and *Colletotrichum panacicola* Uyeda & S. Takim. However, none of these species resembles the new species described in this article.

Although it might appear that the pathogens of *Galinsoga* spp. are not well studied, the number of fungal species reported for this plant is impressive, including some that are considered to be strictly specific to this host plant. *Ascochyta petrakii* Sandu & Mítítíuc (*Didymellaceae*), *Phoma galinsogae* Allesch (*Didymellaceae*), *Septoria galinsogae* Speg. (*Mycosphaerellaceae*), and *Entyloma galinsogae* Syd. & P. Syd. (*Entylomataceae*) were first found and described on *Galinsoga* spp. A specialized fungal pathogen, *Protomyces buerenianus* Buhr (*Taphrinaceae*), which causes swellings and galls on leaves and stems of *G. parviflora*, was described in Germany; later, it was also found in Poland and Slovakia [29,30].

However, *P. buerenianus* is most likely to be a synonym of *Protomyces wodziczkoi* Szulcz, described later [31]. The listed species have *Galinsoga* spp. as the sole host plant; however, their taxonomic status is very precarious as most of them are rarely mentioned and none were sequenced. *Galinsoga* spp. can be infected by powdery mildews caused by *Golovinomyces cichoracearum* (DC.) V.P. Heluta, *Neoerysiphe cumminsiana* (U. Braun) U. Braun, *Sphaerotheca fuliginea* var. *galinsogae* Y.S. Paul & J. Pal, *Oidium* sp., etc. [32–35]. Non-specialized species from *Alternaria* Nees, *Fusarium* Link, and *Sclerotinia* Fuckel have also been recorded on this host plant [36–39]. This list is not limited to Ascomycota, as *Rhizoctonia* DC. sp. and rusts (Basidiomycota), as well as fungus-like organisms, have also been reported in association with *Galinsoga* spp. [38,40,41].

The first evidence of a *Colletotrichum* association with *Galinsoga* spp. comes from the specimen (LEP 135109) of infected *G. parviflora* obtained in 1993 from Golubovka (Ukraine). At that time, the fungus isolated from this specimen was identified as *C. gloeosporioides* based only on morphology [37]. Unfortunately, it is now not possible to molecularly confirm the species for this record as there is no extant culture.

Subsequently, further specimens of *G. parviflora* with anthracnose symptoms were collected, and more fungal strains of *Colletotrichum* spp. were obtained. Further multilocus molecular analysis has made it possible to identify all these strains. In a previous study, a single strain (MF-13.3) was identified as *Colletotrichum coccodes* (Wallr.) S. Hughes [6]. Three other strains were identified and are described here as a new species, *C. galinsogae* sp. nov. It is important to note that both *C. coccodes* (MF-13.3) and *C. galinsogae* sp. nov. (MF-13.2) were isolated from the same specimen (LEP 135112).

A recent study [5] has reported another *Colletotrichum* sp. associated with *Galinsoga* spp., viz. *Colletotrichum fructicola* Prihast., L. Cai & K.D. Hyde (*C. gloeosporioides* species complex) in *Galinsoga quadriradiata* Ruiz & Pav. (using the synonymous name *Galinsoga ciliata* (Raf.) S.F.Blake). To sum up, three *Colletotrichum* species from different species complexes are known to be associated with *Galinsoga* spp. These species are *C. coccodes* (*C. coccodes* species complex), *C. fructicola* (*C. gloeosporioides* species complex), and the species described here, *C. galinsogae* sp. nov. (*C. destructivum* species complex).

The discovery of *C. galinsogae* sp. nov. prompts questions about its origin. This fungus has only been isolated from *G. parviflora* and has not been found on any other plant species, suggesting its strict specialization to this host plant. Given that this plant is non-native to the collection sites and has an origin in Central America, it is plausible that the fungus has the same origin. Therefore, *G. parviflora* might have been introduced to Europe with its specialized fungus Alternatively, this species has another host (as yet unknown) in Europe and has developed pathogenicity against *G. parviflora* more recently. Although this fungus may have always been present in *G. parviflora* growing in the studied locations, the lack of tools to identify it prevented its early recognition by standard morphological approaches. Further data on the biodiversity of *Colletotrichum* spp. associated with *Galinsoga* spp. obtained by multilocus phylogenetic analysis would help clarify the distribution of *C. galinsogae* sp. nov. both globally and in Russia.

**Supplementary Materials:** The following supporting information can be downloaded at: https://www.mdpi.com/article/10.3390/d15070866/s1, Table S1: The list of reference strains used in the current study.

**Author Contributions:** All authors contributed to the conception and design of the study. Materials preparation and data collection were performed by M.G., E.G. and P.G. Molecular work and phylogenetic analyses were performed by M.G. and I.K. The preparation of illustrations was performed by M.G. The first draft of the manuscript was written by I.K., and all authors commented on previous versions of the manuscript. All authors have read and agreed to the published version of the manuscript.

**Funding:** This research was funded by the Russian Science Foundation (RSF), grant number 19-76-30005.

**Institutional Review Board Statement:** Not applicable.

**Informed Consent Statement:** Not applicable.

**Data Availability Statement:** The raw data analyzed during the study are available from the corresponding author on reasonable request.

**Acknowledgments:** We are grateful to Ludmila B. Khlopunova for laboratory assistance and maintaining of the culture collection.

**Conflicts of Interest:** The authors declare no conflict of interest.

## Appendix A

**Table A1.** Origin of *Colletotrichum galinsogae* sp. nov. strains obtained from *Galinsoga parviflora*.

| Strain | Date | Source | Specimen | Location | Coordinates |
|---|---|---|---|---|---|
| MF-13.2 | 17 August 1995 | stem | LEP 135112 | Ukraine, Sumy Oblast, Seredina-Buda district, Golubovka | 52.241277, 33.789763 |
| MF-13.25 | 24 July 2013 | stem | LEP 132984 | Ukraine, Sumy Oblast, Seredina-Buda | 52.194305, 34.040297 |
| MF-13.27 | 9 August 2013 | leaf | LEP 132919 | Russia, Pskov Oblast, Velikie Luki; Maykino | 56.372833, 30.495326 |

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
