# Peer review of "Colletotrichum galinsogae sp. nov. Anthracnose Pathogen of Galinsoga parviflora"

_diversity, doi:10.3390/d15070866_

Round 1

Reviewer 1 Report

I have reviewed the manuscript: the text has a good written, is about an interesting subject and brings new and important information. The molecular study is enough for the fungus Phylogeny, however I have same consideration:

1.       I have a question, why in the pathogenicity test, the authors had not used conidia suspension for inoculation the plants? I think it is not mandatory, but this practice is more usual;

2.       The figures caption need to bring more information (self explanation is mandatory in captions). I have some suggestion in the text.

Author Response

Many thanks to the respected reviewer for the high assessment of our work and useful comments.
1.    I have a question, why in the pathogenicity test, the authors had not used conidia suspension for inoculation the plants? I think it is not mandatory, but this practice is more usual;

In this study, we worked with fungal cultures, some of which were obtained a long time ago, and may have lost the ability to conidiogenesis. Therefore, we unified the experiment by using a mycelial suspension.

2.    The figures caption need to bring more information (self explanation is mandatory in captions). I have some suggestion in the text.

We have changed the captions to the figures. However, we do not think it is appropriate to use the words "photograph" or "microphotograph" because the scale is indicated everywhere in the figures and it is not quite common practice. All changes were highlighted in the text

Reviewer 2 Report

1 Why you use mycelial suspension not conidia suspension in pathogenicity test?

2 It is better if you show symptom in inoculation.

3 What is Adaxial side and Abaxial side in table 1?

4 The pathogen causes spot on leaves, how it caused 100% necrotic leaf area?

5 Each figure in Figure 2 and 3 was not cited in text, it is difficult to understand for readers.

6 What is relationship between figure 2 and figure 3? It is better to combine them together.

7 Charcteristics of colonies and conidia on different media and time were oberseved, in some part you mentioned cultured for 14 days, but in other part you did not mentioned how long cultured.

Author Response

Many thanks to the reviewer for the time spent with the article and valuable comments, as well as for interesting questions on which we will try to answer below

1 Why you use mycelial suspension not conidia suspension in pathogenicity test?

In this study, we worked with fungal cultures, some of which were obtained a long time ago, and may have lost the ability to conidiogenesis. Therefore, we unified the experiment by using a mycelial suspension.

2 It is better if you show symptom in inoculation.

We have all the photos obtained during the inoculation experiment. Nevertheless, we believe that they are of insufficient quality for publication. I am sending one of them as an example (see below)

3 What is Adaxial side and Abaxial side in table 1?

"adaxial" and "abaxial" are botanical and zoological terms. The first means the surface facing the sun, the second the opposite side.

4 The pathogen causes spot on leaves, how it caused 100% necrotic leaf area?

The pathogen was able to form 100 % necrotic leaf area on seedlings. 100 % means that the leaf was completely dead.

5 Each figure in Figure 2 and 3 was not cited in text, it is difficult to understand for readers.

Figures 2 and 3 were cited on page 5 string 6. It seems to us that the introduction of a more detailed description of the figures in the text of the protologue may complicate its reading from the abundance of numerical and alphabetic values relating to the sizes of various fungal macro- and micromorphological structures, genbank numbers, coordinates, e

6 What is relationship between figure 2 and figure 3? It is better to combine them together.

Figure 2 corresponds to morphological characteristics found on herbarium specimen

Figure 3 corresponds to observed cultural characteristics.

Therefore, it is better to keep them separately

7 Charcteristics of colonies and conidia on different media and time were oberseved, in some part you mentioned cultured for 14 days, but in other part you did not mentioned how long cultured.

The necessary changes have been made to the text and highlighted

Reviewer 3 Report

Example of the type of minor editing needed here and there in this manuscript: "This ability could be revive in strains after inoculation host plant and subsequent reisolation". Should be: ...could be revived in strains after inoculation of host plants...

Example of the type of minor editing needed here and there in this manuscript: "This ability could be revive in strains after inoculation host plant and subsequent reisolation". Should be: ...could be revived in strains after inoculation of host plants...

Author Response

Many thanks to the reviewer for his work on the text and valuable comments

1.Here and there in this manuscript: "This ability could be revive in strains after inoculation host plant and subsequent reisolation". Should be: ...could be revived in strains after inoculation of host plants...

Done. All changes were highlighted in the text.